# In Silico Plasma Protein Binding Studies of Selected Group of Drugs Using TLC and HPLC Retention Data

**DOI:** 10.3390/ph14030202

**Published:** 2021-02-28

**Authors:** Karolina Wanat, Grażyna Żydek, Adam Hekner, Elżbieta Brzezińska

**Affiliations:** Department of Analytical Chemistry, Faculty of Pharmacy, Medical University of Lodz, 92-216 Lodz, Poland; grazyna.zydek@umed.lodz.pl (G.Ż.); adam.hekner@umed.lodz.pl (A.H.); elzbieta.brzezinska@umed.lodz.pl (E.B.)

**Keywords:** protein binding, human serum albumin, statistical modeling, chromatographic data, QSPR, thin layer chromatography, high performance liquid chromatography

## Abstract

Plasma protein binding is an important determinant of the pharmacokinetic properties of chemical compounds in living organisms. The aim of the present study was to determine the index of protein binding affinity based on chromatographic experiments. The question is which chromatographic environment will best mimic the drug–protein binding conditions. Retention data from normal phase thin-layer liquid chromatography (NP TLC), reversed phase (RP) TLC and HPLC chromatography experiments with 129 active pharmaceutical ingredients (APIs) were collected. The stationary phase of the TLC plates was modified with protein and the HPLC column was filled with immobilized human serum albumin. In both chromatographic methods, the mobile phase was based on a buffer with a pH of 7.4 to mimic physiological conditions. Chemometric analyses were performed to compare multiple linear regression models (MLRs) with retention data, using protein binding values as the dependent variable. In the course of the analysis, APIs were divided into acidic, basic and neutral groups, and separate models were created for each group. The MLR models had a coefficient of determination between 0.73 and 0.91, with the highest values from NP TLC data.

## 1. Introduction

Protein binding (PB) is an important consideration in the design of new drug substances. Only the free form of the drug is capable of pharmacodynamic action and passing through biological barriers. However, as proteins are widely distributed throughout the human body, it seems impossible to prevent them from interacting with medications [1,2]. For this reason, a number of in vitro and in vivo methods have been developed to analyze PB. Of these, chemometric methods, i.e., those that use measurement data to predict the chemical, physical or biological properties of compounds, are gaining increasing importance [3,4,5], mainly due to their relative simplicity and low cost.

Binding with proteins occurs most often as a result of hydrophobic, van der Waals and electrostatic interactions. The most important drug binding proteins are human serum albumin (HSA), alpha-1-acid glycoprotein (AGP), transferrin, transcobalamin, thyroglobulin, haptoglobin, corticosteroid binding protein, lipoproteins and immunoglobulins [6]. HSA is responsible for the majority of PB in human plasma [7,8], and is mainly responsible for the binding of neutral or acidic hydrophobic compounds [9].

The aim of this study is to find the best chromatographic environment in which the PB conditions of the body can be recreated. We tested a range of chromatographic experiments (normal phase thin-layer liquid chromatography (NP TLC), reverse phase (RP) TLC and HPLC) as indicators of the PB affinity of selected active pharmaceutical ingredients (APIs), with the assumption that results could be used to predict the PB. The successful model should demonstrate the protein binding at the physiological pH of human plasma; therefore, the mobile phase was based on a buffer at pH 7.4. Comparing the effectiveness of these methods and proposing mathematical modifications of the retention values of the studied APIs can be a source of convenient quantitative parameters. The chromatographic data obtained in the subsequent experiments were compared with the PB values from the literature [10]. 

In the HPLC experiment, a commercial column was used, the packing of which was immobilized with HSA (HPLC_HSA_). The values of the retention factor (k) were collected, along with a more useful derivative thereof: log k.

Thin-layer chromatography (TLC) was performed in normal and reverse phase (NP and RP) systems. The stationary phase surface-modifying protein was bovine serum albumin (BSA). This protein is often used as an effective replacement for human plasma albumin when studying the affinity of drugs to protein [11,12]. Such a substrate should also offer good adhesion of the protein modifier, even coverage of the plate surface and be as immobile as possible under chromatographic conditions.

Chromatographic data were later transformed using physicochemical properties such as lipophilicity or polar surfae areas. All descriptors used in the multiple linear regression are listed in Table 1. 

## 2. Results

### 2.1. HPLC_HSA_ Column Chromatography Model

In this study, we examined the retention factor (log k) and all derivatives related to the physicochemical properties of the tested APIs. A correlation matrix of all chromatographic variables was created and the level of plasma protein binding (PB_abn_) was calculated. In this group, the drugs of bases (b), acids (a) and neutral (n) character were tested. The log k values demonstrated a significant correlation with PB_abn_: R = 0.56 (number of examined cases: n_abn_ = 128). The interactions between the remaining log k derivatives were observed in terms of their mutual correlations. The introduction of all unrelated variables (log k, log k/B2 and log k/PSA) into the multiple linear regression (MLR) analysis did not increase the correlation with the PB_abn_ (R = 0.57, n_abn_ = 129). This result is much less satisfactory than expected.

For the acidic APIs (a), the analysis showed significant dependencies. The values of log k and its derivatives were found to be directly proportional to the level of protein binding. The correlations were high, especially for log k (R = 0.58) and log k/PB (R = 0.63) (n_a_ = 28). It was not possible to construct a single mathematical model built on more than one independent variable due to the very strong relationships between the variables. The acidic group was also characterized by a small number of cases, and the significant correlations of chromatographic variables observed here may depend on the polar nature of acidic drugs (log k and log k/PSA).

The group of neutral APIs (n) consisted of 63 cases. The correlation matrix showed no mutual associations between the chromatographic variables (except log k and log k/PB) but showed a fairly high correlation between log k and PB_n_ (R = 0.58, n_n_ = 63). MLR analysis, performed using the progressive stepwise method, did not improve the result. The combined cases (a) and (n) constituted a larger group, with the number of cases n_an_ = 91. Such structures were most often bound by human serum albumin. This increase in the number of cases did not reduce the correlation between PB_an_ and the independent variable log k: R = 0.57 (n_an_ = 91), and even increased it very slightly. The dependence of PB_an_ on log k/PB: R = 0.57 (n_an_ = 91) also increased. Log and log k/PB remained strongly related, and the subsequent independent variables did not show mutual correlations. This allowed for an attempt to establish multiple regression. Two independent variables appeared in the resulting MLR equation: log k, log k/B2. This model explained 33% of the total variability of the PB_an_ index (R = 0.58).

Chromatographic data from the HPLC_HSA_ column of base APIs (b) were also analyzed. This group consisted of 34 cases (n_b_ = 34). The relationship between the chromatographic data and the level of protein binding was similar to the previous groups, but it seemed more favorable. All independent variables were directly proportional to the PB_b_ value. As no significant correlations were found between the independent variables, they appeared useful. This time, the correlation between log k/PB and PB_b_ was very weak and amounted to only 0.22. The correlation between PB_b_ and the independent variable log k increased to R = 0.63 (n_b_ = 34). This single variable explained 40% of the total variance of PB_b_. The MLR equation, containing log k and log k/PSA, explained 41% of the variation in PB_b_ (R = 0.64, n_b_ = 34). Increasing the group of bases (b) with neutral drugs (n) caused another change in the observed relationships. The number of cases increased to 100, but the level of correlation with the PB_bn_ index did not change significantly. The chromatographic data and their derivatives were not related to each other. The MLR model built with all independent variables was no better than the correlation between PB_bn_ and log k. The equation was statistically significant, but the result was not satisfactory (R = 0.57, n_bn_ = 100).

### 2.2. NP and RP Thin-Layer Chromatography Models

The prognostic value of the obtained parameters (variables) were tested in the NP system. The analysis included 129 acid, base and neutral APIs in total, and six independent variables. The correlation matrix indicated no visible relationships between the independent variables. Hence, an MLR mathematical model was created to test the dependence of the PB on the established chromatographic parameters of NP-TLC. MLR analysis was performed using the stepwise method. The model included the following variables: NP, NP/C, NP/PSA and NP/PB. All variables were statistically significant. The correlation for the dependent variable was high (R = 0.90, n_abn_ = 129). The model explained 81% of the total variability of PB_abn_—Equation (1)—in the group of all tested drugs (a, b, n).
PB_abn_ = 1.23(±0.28) − 0.27(±0.01) NP/PB + 0.58(±0.06) NP + 0.02(±0.008) NP/PSA − 0.61(±0.29) NP/C
R= 0.90; R^2^ = 0.81 F(4,124) = 136.12; *p* < 0.0000; s = 0.12549; n_abn_ = 129(1)
Q^2^_LOO_ = 0.70, SDEP= 0.1569, PRESS =3.1970, S_PRESS_ = 0.1568, Q^2^_LMO_ = 0.63

A good representation of drug protein binding can be seen in the scatter plot of predicted PB versus observed PB given below (Figure 1).

Chromatographic data, in the form of R_M_ values and their derivatives, showed lower correlation with PB_abn_. The most important parameter was R_M_NP (R = 0.26, n_abn_ = 129), which was inversely proportional to protein binding. The attempt to use all independent variables for the MLR analysis confirmed a significantly worse fit of PB_abn_ and R_M_NP − R = 0.41, n_abn_ = 129.

Further analyses were performed using the cases (a), (b) and (n); the results for each group are presented in Table 2. 

Very good results were obtained for the cases from the group of acids (a), with n_a_ = 29. The correlation matrix showed a good fit of all variables to PB_a_ and no mutual relationship was found between the independent variables. The NP correlation value increased to R = 0.48. The mathematical model based on the MLR stepwise analysis explained 91% of the total variability in PB in the (a) group. The equation was statistically significant and despite the small number of cases, the result can be considered very good: R = 0.95, n_a_ = 29. 

Good results were also recorded for basic drugs (b). MLR analysis explained over 91% of the total variation in PB_b_. The equation was statistically significant and despite a small group of cases, the result can be considered very good: R = 0.96, n_b_ = 34. Scatter plots for PB_a_, PB_b_ and PB_n_ are given in Appendix B (Figure A1, Figure A2 and Figure A3). Results with R_M_ parameters for (a), (b) and (n) were significantly lower and they are not presented here.

MLR analysis was then performed in the combined groups (ab), (an) and (bn) (Table 3). Alkaline and neutral (bn) compounds account for the largest proportion of all cases, i.e., 100 cases. A mathematical model (Figure 2) was developed for this large group of drugs, which was found to account for 79% of protein binding (R = 0.89, n_bn_ = 100). Thus, the obtained result had a very similar prognostic value to the study of the entire group of cases together. Regarding the calculated R_M_ parameters for this group, the highest correlation with the PB_bn_ index was obtained for R_M_NP/C (R = 0.24, n_bn_ = 100). After introducing the variables R_M_NP/C, R_M_NP/PSA and R_M_NP/logP, the mathematical model explained only 11% of the variability of PB (R = 0.33, n_bn_ = 100).

The chemicals tested in the two experiments described above were subjected to RP TLC analysis. All cases examined in this experiment comprised a group of 129 compounds. The correlation matrix revealed no relationship between the chromatographic variables of R_f_. The independent variables in the model were not related to each other. The model (Equation (2); Figure 3) explained 76% of the variability of the PB index. However, the model yielded a worse result than in the case of the BSA-modified NP stationary phase chromatographic experiment (see Equation (1)). Furthermore, protein binding studies for the entire group of cases with R_M_RP variables were not conclusive.
PB_abn_ = 0.76(±0.05) + 0.41(±0.07) RP − 0.22(±0.01) RP/PB + 0.004(±0.002) RP/B2
R= 0.87; R^2^ = 0.76; F(3,125) = 132.78; *p* < 0.0000; s = 0.14191; n_abn_ = 129(2)
Q^2^_LOO_ = 0.72, SDEP = 0.1514, PRESS = 2.9392, S_PRESS_ = 0.1509, Q^2^_LMO_ = 0.72

As before, the statistical analysis was repeated for all three types of cases (a, b, n) and in mixed groups (an, bn, ab). The best MLR results—achieved for (a) and (ab)—are gathered in Table 4.

## 3. Discussion

The log k variable appears to play a key role in predicting drug-protein binding based on HPLC_HSA_ chromatographic data. This parameter is strongly and directly proportionally correlated with PB. This correlation was observed in the acidic, neutral and basic groups, and in all combinations between them. However, the degree of the correlation changes very slightly for groups of different sizes, with R values within 0.56–0.63 for the number of cases between 27 and 128. The highest value (0.63) was observed for basic drugs (n_b_ = 34). Analysis of the HPLC_HSA_ data also showed that the creation of an independent chromatographic variable containing a PB value (log k/PB) did not directly increase the correlation with the dependent variable PB in any group of cases.

For all groups of cases examined by NP TLC, the chromatographic parameters describe the ability of drugs to bind to proteins, both high (for the observation of R_f_ data) and low (for R_M_). This ability was similar for all groups, regardless of significant differences in structure, acid-base character and the group size, i.e., 129, 100, 95, 63, 34 or 29. The correlation coefficient ranged from 0.89 to 0.96. Mathematical models with the participation of R_f_ variables explained 79–91% of the variability of PB in groups; these most often contained NP, NP/B2 and NP/PB as independent variables. BSA-modified NP TLC analysis appears to provide data (R_f_ and derivatives) on protein binding for any drug class. R_M_ variables, unfortunately, can be considered of little use in predicting the level of PB. Interestingly, the scatter plot of the observed and predicted PB values highlighted three groups of cases: one with a high level of plasma protein binding, i.e., from 85% to 100%; a medium level of binding, i.e., from 25% to 85%; and a low level, i.e., from 0% to 25%. Reducing the affinity to proteins is also associated with the distance between their predicted values and the trend line.

In the RP TLC analysis, the R_f_-related variables were more efficient. These were found to highly correlated with the dependent variable PB_b_−R = 0.86 (n_b_ = 34), but the equation was not statistically significant. There were three independent variables in the model: RP/C, RP/PSA and RP/PB. The intercept of this equation was also not statistically significant, which proves that such a model cannot be applied. The group of neutral compounds was twice the size (n_n_ = 66); therefore, it was possible to construct an MLR model for the PB_n_ using three RP variables RP/PB, RP/C and RP/logP. The obtained result was good: R = 0.92, coefficient of determination R^2^ = 0.84. Unfortunately, the intercept in the resulting equation had the wrong parameters, it was not statistically significant and the standard error was 100 times its value. Therefore it cannot be used to predict the dependent variable.

After the R_M_RP dataset and their derivatives were introduced into the analysis of PB_a_, PB_b_ and PB_n_, numerous interrelationships of the independent variables appeared. This significantly reduced the possibilities of the analysis.

When the acidic and neutral drug groups were combined (PB_an_), the intercept in the resulting equation had the wrong parameters, was not statistically significant and the standard error was equal to its value. Such an equation cannot be used to predict the dependent variable. The use of R_M_RP data yielded a model with a lower value. The model included only the R_M_RP/PSA and R_M_RP2/PB variables. The intercept was not statistically significant, and the equation explained 45% of the total variability of the PB_an_.

The scatter plots (from the R_f_ variables) in each of the independent case groups—a, b, and n—in the TLC experiments show a very similar, non-linear shape of the case distribution. The dispersion characteristic is also a representation of the dispersion of all considered cases together (see Figure 1). This distribution causes poorer predictions for drugs with moderate PB values, between 0.4 and 0.7. This suggests that the dependent variable can be mathematically transformed to better fit the model.

An accurate analytical model of the drug in a living organism allows APIs of different structure and properties to be tested. The result should also be resistant to the size of the studied group of cases. A comparison of subsequent analytical models is presented below (Table 5). The research included the entire group of 129 originally tested APIs, (abn), (a), (b), (n), plus 38 external cases (structures and data available in the Appendix A). APIs from the external group were also divided into acidic (n_a_ = 6), basic (n_b_ = 16) and neutral groups (n_n_ = 16) and were subjected to chromatographic experiments under the same conditions.

Fluctuations in the size of the studied groups of cases did not affect the correlations with the dependent variable. All chromatographic experiments with plasma proteins as part of the stationary phase yielded analytical models that were resistant to changes in group size. The best correlation results were obtained with the HPLC_HSA_ experiment. Such a result is obvious because the data of the dependent variable (PB) obtained from the literature concern the binding of drugs to human serum albumin, which is part of the construction of the stationary phase of the HSA column. The remaining experiments were performed with bovine serum albumin. The problem with HPLC data are the poor results of the MLR models, which cannot be used in predicting PB.

## 4. Materials and Methods

The chromatographic experiments and the methods of data collection are described in detail in Appendix C.

### Statistical Modeling and Stepwise Multiple Linear Regression

The goal of multiple linear regression is to quantify the relationship between multiple independent (explanatory) variables and the dependent variable. The protein binding (PB) values were used as dependent variables [10]. Physicochemical properties PSA and log P were calculated in HyperChem (HyperChem for Windows Release 7.02, HyperCube Inc, 2002) and later used to modify the retention data. Computational descriptor B2, which describes the bioavailability in the central nervous system, was calculated from the equation log bb = 0.547 − 0.016 PSA [13]. Acid-base properties were collected from the literature [14]. 

MLR was performed in stepwise mode, in STATISTICA 13.1 (TIBCO Software Inc.) software. Validation of regression models was performed using general internal cross-validation procedures: “leave-one-out” (LOO) and “leave-many-out” (LMO). In the LOO validation, one case is removed from the dataset and used to verify the model built with the remaining elements; the procedure is then repeated with other elements. In the LMO approach, the dataset is divided into two subsets (25% and 75%), used for model construction and its evaluation, respectively. The prediction power of the models was estimated using the cross-validated squared correlation coefficient (Q^2^_LOO_), predicted residual sum of squares (PRESS), standard deviation based on PRESS (S_PRESS_) and standard deviation of the error of prediction (SDEP). The suggested criteria for predicting the accuracy of MLR models [15] are R^2^ > 0.6 and Q^2^
_LO(M)O_ > 0.5; R^2^ ≥ Q^2^
_LO(M)O_ and Q^2^
_LOO_ ≈ Q^2^
_LMO_.

## 5. Conclusions

The great influence of drug–protein binding on pharmacotherapy has resulted in the development of many methods for its evaluation and determination. Such analytical models for the investigation of drug–protein binding in the body can be based on simple laboratory analyses such as TLC or HPLC.

Our findings demonstrate the value of chromatographic data in plasma protein binding studies in general, and for acidic (a), neutral (n) and basic (b) compounds. The correlations observed for the PB_abn_ models, i.e., for all tested compounds together, did not differ significantly from those observed for narrowed groups (a), (b) and (n), despite a much larger number of cases. Additionally, the analytical models were found to be resistant to random effects, which can be noticed while increasing the number of cases in all groups (a, b, n, abn). Interestingly, the drug lipophilicity value (log P) was very small. Log P is considered to be one of the most important determinants of protein binding; however, it was found to be important only for basic compounds (b), which can demonstrate unspecific BSA binding. Of greater importance is the PSA value, related to the ionization of the compounds, and the B2 value, describing penetration into the central nervous system.

Chromatographic data can be important independent variables in mathematical models, especially in combination with physicochemical drug descriptors relevant to PB. However, the main aim of this work was to determine the predicted levels of drug protein binding by comparing different affinity chromatography environments. The best analytical models were obtained using NP TLC with BSA-modified plates, with R_f_ values and their derivatives. The results were significantly better than those obtained from HPLC using a commercial column with immobilized HSA. In addition, NP TLC, with its relative simplicity and low cost of analysis, can be a useful method for protein binding analysis. In addition, stationary phase modification may provide new options for TLC experiments.

## Figures and Tables

**Figure 1 pharmaceuticals-14-00202-f001:**
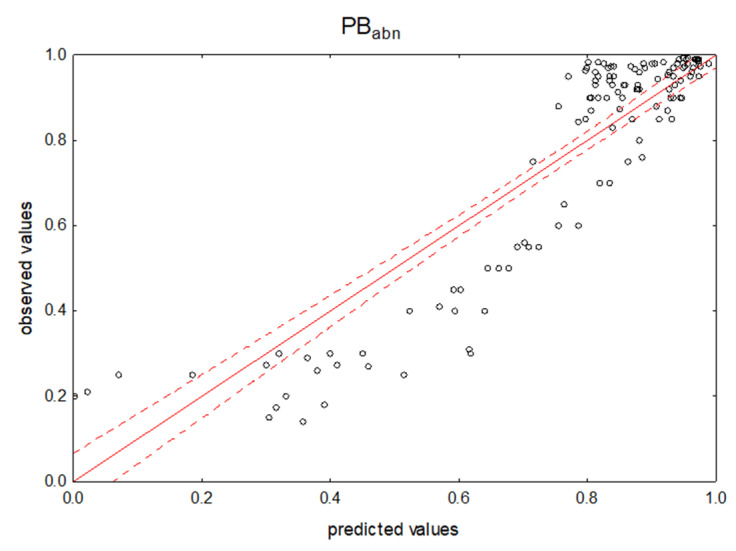
Predicted vs. observed values for the multiple linear regression (MLR) model using NP TLC retention data. Dependent variable: PB_abn_.

**Figure 2 pharmaceuticals-14-00202-f002:**
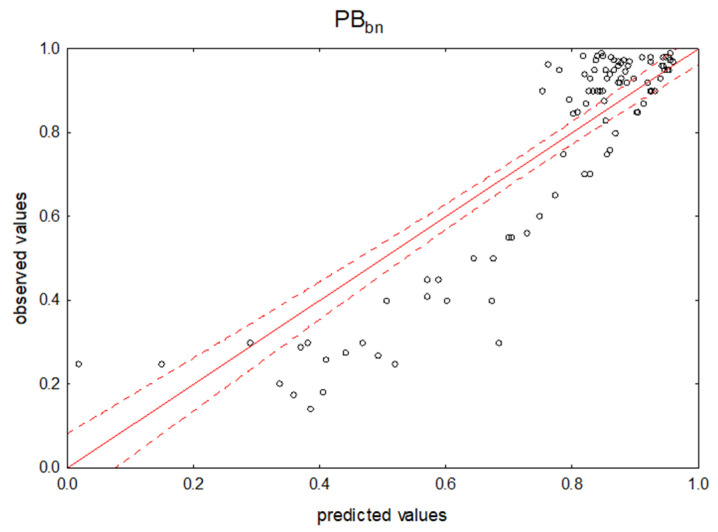
Predicted vs. observed values for the MLR model using NP TLC retention data. Dependent variable: PB_bn_.

**Figure 3 pharmaceuticals-14-00202-f003:**
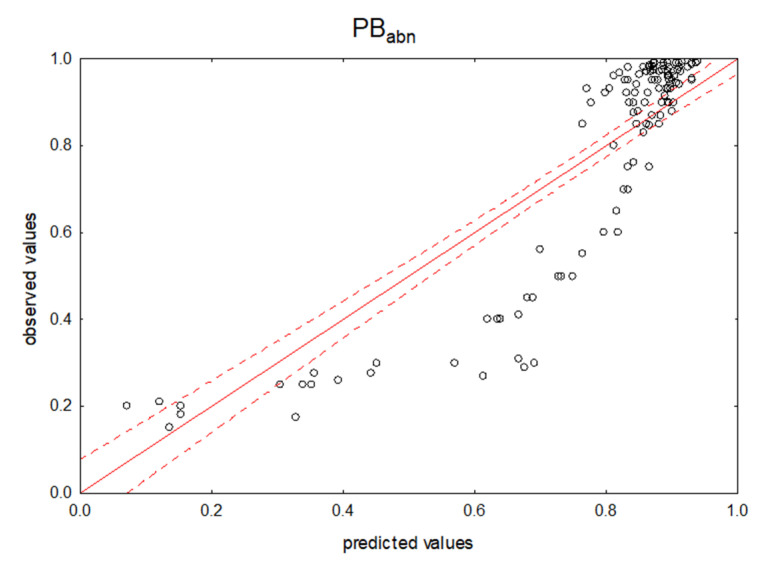
Predicted vs. observed values for MLR model using RP TLC retention data. Dependent variable: PB_abn_.

**Table 1 pharmaceuticals-14-00202-t001:** Chromatographic parameters from normal phase/reverse phase thin-layer liquid chromatography (NP/RP TLC) and HPLC_HSA_ experiments and their derivatives used in the analysis of analytical models as independent variables.

Chromatographic Parameter or the Derivative	Description
NP; RP	The R_f_, obtained in TLC chromatography on the bovine serum albumin (BSA)-impregnated plates, in normal and reversed mode, respectively
NP/C; RP/C	The R_f_ from BSA-impregnated NP or RP plates/the R_f_ from a clean plate
NP/PSA; RP/PSA	The R_f_ from BSA-impregnated NP or RP plate/polar surface area
NP/PB; RP/PB	The R_f_ from BSA-impregnated NP or RP plate/protein binding value
NP/logP; RP/logP	The R_f_ from BSA-impregnated NP or RP plate/partition coefficient
NP/B2; RP/B2	The R_f_ from BSA-impregnated NP or RP plate/computational parameter B2, describes the bioavailability in the central nervous system
R_M_NP; R_M_RP	The R_M_, obtained in TLC chromatography on the BSA-impregnated plates, in normal and reversed mode, respectively
R_M_NP/C; R_M_RP/C	The R_M_ from BSA-impregnated NP or RP plate/the R_M_ from a clean plate
R_M_NP/PSA; R_M_RP/PSA	The R_M_ from BSA-impregnated NP or RP plate/polar surface area
R_M_NP/PB; R_M_RP/PB	The R_M_ from BSA-impregnated NP or RP plate/protein binding value
R_M_NP/logP; R_M_RP/logP	The R_M_ from BSA-impregnated NP or RP plate/partition coefficient
R_M_NP/B2; R_M_RP/B2	The R_M_ from BSA-impregnated NP or RP plate/computational parameter B2, describes the bioavailability in the central nervous system
log k	logarithm of retention factor from HPLC_HSA_
log k/PSA	logarithm of retention factor from HPLC_HSA_/polar surface area
log k/PB	logarithm of retention factor from HPLC_HSA_/protein binding value
log k/logP	logarithm of retention factor from HPLC_HSA_/partition coefficient
log k/B2	logarithm of retention factor from HPLC_HSA_/computational parameter B2, describes the bioavailability in the central nervous system

**Table 2 pharmaceuticals-14-00202-t002:** MLR models for acidic (a), basic (b) and neutral (n) active pharmaceutical ingredients (APIs) using NP TLC retention data.

No of Cases	Stepwise Multivariate Linear Regression Model	MLR Model Parameters
a n_a_ = 29	PB_a_ = −2.99(±1.23) + 0.48(±0.10) NP + 3.53(±1.26) NP/C + 0.15(±0.04) NP/PSA − 0.21(±0.02) NP/PB	R = 0.95; R^2^ = 0.91; F(4,24) = 57.434; *p* < 0.0000; s = 0.09925 Q^2^_LOO_ = 0.81, SDEP = 0.1308, PRESS = 0.5376, S_PRESS_ = 0.1362, Q^2^_LMO_ = 0.81
b n_b_ = 34	PB_b_ = 0.76(±0.054) + 0.88(±0.11) NP − 0.54(±0.03) NP/PB	R = 0.96; R^2^ = 0.91; F(2,31) = 160.74; *p* <0.0000; s = 0.09437 Q^2^_LOO_ = 0.88, SDEP= 0.1063, PRESS = 0.3636, S_PRESS_ = 0.1034, Q^2^_LMO_ = 0.86
n n_n_ = 66	PB_n_ = 0.80(±0.05) + 0.44(±0.06) NP − 0.29(±0.02) NP/PB	R= 0.90; R^2^ = 0.81; F(2,64) = 133.29; *p* < 0.0000; s = 0.11958 Q^2^_LOO_ = 0.77, SDEP = 0.1298, PRESS = 1.0890, S_PRESS_ = 0.1285, Q^2^_LMO_ = 0.76

**Table 3 pharmaceuticals-14-00202-t003:** MLR models for combined groups of APIs: (bn), (an) and (ab) using NP TLC retention data.

No of Cases	Stepwise Multivariate Linear Regression Model	MLR Model Parameters
bn n_bn_ = 100	PB_bn_ = 0.74(±0.037) + 0.56(±0.06) NP − 0.34(±0.02) NP/PB	R= 0.88; R^2^ = 0.78, F(2,97) = 174.29; *p* < 0.0000; s = 0.1339 Q^2^_LOO_ = 0.75, SDEP = 0.1420, PRESS = 1.9875, S_PRESS_ = 0.1410, Q^2^_LMO_ = 0.75
an n_an_ = 95	PB_an_ = 0.74(±0.05) + 0.49(±0.05) NP − 0.26(±0.01) NP/PB + 0.01(±0.00) NP/B2	R= 0.89; R^2^ = 0.76, F(3,91) = 111.21; *p* <0.0000; s = 0.13042 Q^2^_LOO_ = 0.70, SDEP = 0.1504, PRESS =2.2123, S_PRESS_ = 0.1526, Q^2^_LMO_ = 0.69
ab n_ab_ = 63	PB_ab_ = 1.39(±0.41) + 0.80(±0.09) NP − 0.30(±0.02) NP/PB + 0.01(±0.00) NP/B2 + 0.03(±0.01) NP/PSA − 0.89(±0.43) NP/C	R= 0.89; R^2^ = 0.80, F(5,57) = 44.235; *p* < 0.0000; s = 0.14432 Q^2^_LOO_ = 0.63, SDEP = 0.1883, PRESS = 2.6723, S_PRESS_ = 0.2060, Q^2^_LMO_ = 0.63

**Table 4 pharmaceuticals-14-00202-t004:** MLR models for acidic drugs (a) and combined group of APIs (ab) using RP TLC retention data.

No of Cases	Stepwise Multivariate Linear Regression Model		MLR Model Parameters
a n_a_ = 29	PB_a_ = 0.68(±0.09) + 0.43(±0.12) RP − 0.23(±0.02) RP/PB + 0.01(±0.00) RP/B2 + 0.10(±0.04) RP/PSA	R= 0.93; R^2^ = 0.86; F(4,24) = 36.4743; *p* < 0.0000; s = 0.1213 Q^2^_LOO_ = 0.74, SDEP = 0.1541, PRESS = 0.8425, S_PRESS_ = 0.1704, Q^2^_LMO_ = 0.74	
ab n_ab_ = 63	PB_ab_ = 0.71(±0.07) + 0.46(±0.10) RP − 0.21(±0.02) RP/PB	R= 0.85; R^2^ = 0.73; F(2,60) = 80.471; *p* < 0.00000; s = 0.16193 Q^2^_LOO_ = 0.67, SDEP = 0.1772, PRESS = 1.9589, S_PRESS_ = 0.1763, Q^2^_LMO_ = 0.62	

**Table 5 pharmaceuticals-14-00202-t005:** Correlation of chromatographic parameters (log k and R_f_ from NP and RP TLC) with protein binding values. Relations were made for originally tested APIs (129 drugs) and for enlarged groups with external cases (+38 drugs).

Group		n	log k ***	NP ***	RP ***
a	Tested *	29	0.63	0.48	0.09
	+external **	35	0.49	0.56	0.16
n	tested	65	0.60	0.09	−0.32
	+external	80	0.67	0.07	−0.24
b	tested	34	0.67	0.22	−0.37
	+external	50	0.63	0.03	−0.45
bn	tested	99	0.56	0.17	−0.28
	+external	131	0.63	0.09	−0.28
an	tested	94	0.59	0.21	−0.15
	+external	115	0.58	0.22	−0.09
ab	tested	63	0.61	0.37	−0.05
	+external	85	0.54	0.31	−0.01
abn	tested	129	0.58	0.24	−0.16
	+external	165	0.57	0.20	−0.16

*—originally tested APIs. **—originally tested APIs + external drugs. ***—correlation with PB.

## Data Availability

Not applicable.

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
