# Peer review of "In Silico Plasma Protein Binding Studies of Selected Group of Drugs Using TLC and HPLC Retention Data"

_pharmaceuticals, 2021, doi:10.3390/ph14030202_

Round 1
Reviewer 1 Report
- please restate the abstract part.
- 1) reduce introduction and method parts
- 2) summarize the results
- 3) write down the conclusion
- seperate and describe the conlusions in the the text
Author Response
The abstract was re-written, this time the aim of the study is more defined.
In the introduction, I made a few changes (in the revised version of the manuscript: lines 40-47, ), which I consider necessary.
The materials and methods part describing the chromatographic experiments in detail have been moved to Appendix B because Reviewer # 4 suggested an improvement to this section.
Part of the introduction (lines 56-72 from the orginal version) have been moved to the Appendix B as well, so this section is significantly shorter.
A final part was also written (lines 254-276), summarizing the results and formulating conclusions.
Reviewer 2 Report
in this paper, several chromatography data was used and designed as an index of protein binding affinity. High performance liquid chromatography with HSA column, Thin-layer chromatography in NP and RP2 system etc
strenght: Chemometric analyses were performed to compare multiple linear regression models (MLR) with retention data and protein binding values as the dependent variable. In the course of the analysis, APIs were divided into acidic, basic and neutral groups, and separate models were created for each of them. weakness: please check carefully english language. it would be good to analyse quantitatively by hptlc
Author Response
The English language was checked by the Translation Office of the Medical University of Lodz after receiving the Reviewers reports. They proposed some changes to the overall sentence structure to make the text more "legible".
In our study, we focus on relatively cheap methods such as simple TLC, but our densitometer and sample applicator are also compatible with HPTLC plates as well, we may consider this option in the near future
Reviewer 3 Report
In the paper entitled “ in silico plasma protein binding studies of selected group of drugs using TLC and HPLC retention data”, Wanat et al. present a combined experimental and theoretical study. It is hard to infer the main purpose of this study; however, I assume it aims to deepen our understanding of the plasma protein binding of drugs. The paper is poorly written; making it hard to evaluate the appropriateness of methods and the solidity of the findings. For example, there is no mention of control experiment for the chromatography experiments. As a negative control, all the baseline API retentions should be measured in the absence of protein. Although plasma protein binding is an important factor in the development of pharmaceutics, the findings of this paper are not of an immediate impact for the readership of the journal Pharmaceuticals. Considering these, I can not recommend publication of this manuscript. I suggest authors to extensively edit the content, language and organization of the paper and submit it to a more relevant journal.
Author Response
The aim of the study which is to find the best chromatographic environment where the protein binding conditions can be recreated has been further defined in the revised manuscript (in the Abstract, in the main text: lines 40-47, 255-257 and 269-271).
The control of TLC experiment was described in detail in the original manuscript, lines 60-66:
“The mobility of the APIs was initially determined on NP and RP2 plates without the participation of the protein modifier. This experiment was called the control (C). The obtained retardation factor (Rf) values made it possible to determine the composition of the mobile phase and to observe the specific effect of the protein modifier on the mobility of the analytes under these conditions. The models also used the Rf/C parameter - the ratio of the Rf value on the plates modified with albumin to the Rf value on the plates without albumin. This parameter better reflects the effect of the modifier on the chromatography of each API.”
In our models we also used descriptors based on this chromatographic value: NP/C or RP/C which are the ratio between The Rf from BSA impregnated NP or RP plate and the Rf from a clean plate. In the revised manuscript, above-mentioned descriptions were moved into the Appendix B (now lines 334-340)
As for the text structure, I contacted the Translation Office of the Medical University of Lodz after receiving the opinion of the Reviewers. They proposed some changes to the overall sentence structure to make the text more readable and understandable, while the topic might be difficult for non-experts to understand.
The manuscript was sent to Pharmaceuticals after receiving a written invitation to participate in a Special Issue on the “6th International Electronic Conference on Medicinal Chemistry”. Part of the study was presented at the conference, and the article will be a continuation and development of the topic. In this case, I would prefer the Editors of the Special Issue to decide whether this journal is appropriate for such article or not.
Reviewer 4 Report
The manuscript focused on the prediction of protein binding ability with pharmaceuticals predicted using HPLC and TLC retention data. In my opinion, the work should be significantly improved with respect to the data collection, processing and interpretation.
Data collection - the experimental conditions of HPLC should be improved. It is not clear how k values were calculated - the marker of dead volume is missing, temperature of the column is missing, injected volume is missing. The authors should also add at least representative chromatogram to illustrate the method (i.e. into the supporting material).
Data processing and interpretation - most of the correlations, i.e. dependence of observed values on predicted values (figs. 1-3) have shown nonlinear trend with convex-like character. The authors should deal with this within discussion; with fitting of linear line (and calculating Pearson's correlation coefficient), in this case for low observed value, the predicted values are underestimating the situation and vice versa. The discussion should take into account this trend.
Some minor comments:
Why RP2 abbreviation as reversed phase system? RP is usually used.
According to IUPAC change retention coefficient and retention index to retention factor – line 55, 79.
Line 270 – „retention factor Rf“ should read „retardation factor Rf“
Author Response
The Reviewer #1 suggested to shorten the materials and methods section. Reviewer #4 had a different opinion, so I decided to move the part on chromatographic experiments to Appendix B and improve it there. I added more details (injection volume, void time, temperature conditions etc) about HPLC and data collection as requested (lines 327-334), attached sample chromatogram (Figure B1) with description. I also attached a sample chromatogram to the TLC experiment (Figure B2), from a control and impregnated plate.
The Discussion section has also been improved, the RP TLC section has been added (lines 189-206). Table 5 is attached along with a discussion of the stability of the correlation value with increasing number of cases (lines 212-231). The nonlinearity issue was previously mentioned in the Results section, but it was rightly suggested to be included in Discussion (lines 207-211).
A Conclusion section was also written (lines 254-276), summarizing the results and formulating conclusions.
Minor mistakes that were corrected:
retention factor into retardation factor (now in line 342, in the Appendix B)
retention index/coefficient into retention factor (now in lines 50 and 67)
RP2 abbreviation was used because TLC plates were RP-2 type, but to make it more easy to read RP2 was changed into RP. The type of plates are mentioned in the Appendix B, in Materials and methods.
Round 2
Reviewer 4 Report
In the revised version of the manuscript, the authors have significantly improve the quality of the data treatment, presentation and discussion. With respect to the current version, I consider manuscript acceptable for publication.